# Effect of Simultaneous Mechanical and Electrical Stress on the Electrical Performance of Flexible In-Ga-Zn-O Thin-Film Transistors

**DOI:** 10.3390/ma12193248

**Published:** 2019-10-04

**Authors:** Youngjin Seo, Hwan-Seok Jeong, Ha-Yun Jeong, Shinyoung Park, Jun Tae Jang, Sungju Choi, Dong Myong Kim, Sung-Jin Choi, Xiaoshi Jin, Hyuck-In Kwon, Dae Hwan Kim

**Affiliations:** 1School of Electrical Engineering, Kookmin University, Seoul 02707, Korea; ssoocw1535@kookmin.ac.kr (Y.S.); shinyoung94@kookmin.ac.kr (S.P.); jtjang@kookmin.ac.kr (J.T.J.); sungjuchoi@kookmin.ac.kr (S.C.); dmkim@kookmin.ac.kr (D.M.K.); sjchoiee@kookmin.ac.kr (S.-J.C.); 2School of Electrical and Electronics Engineering, Chung-Ang University, Seoul 06974, Korea; hwanseok518@cau.ac.kr (H.-S.J.); seonmmmn9@cau.ac.kr (H.-Y.J.); 3School of Information Science and Engineering, Shenyang University of Technology, Shenyang 110870, China; xsjin@live.cn

**Keywords:** Flexible IGZO TFTs, Al_2_O_3_ gate dielectric, simultaneous mechanical and electrical stress, hydrogen

## Abstract

We investigated the effect of simultaneous mechanical and electrical stress on the electrical characteristics of flexible indium-gallium-zinc oxide (IGZO) thin-film transistors (TFTs). The IGZO TFTs exhibited a threshold voltage shift (*∆V*_TH_) under an application of positive-bias-stress (PBS), with a turnaround behavior from the positive *∆V*_TH_ to the negative *∆V*_TH_ with an increase in the PBS application time, whether a mechanical stress is applied or not. However, the magnitudes of PBS-induced *∆V*_TH_ in both the positive and negative directions exhibited significantly larger values when a flexible IGZO TFT was under mechanical-bending stress than when it was at the flat state. The observed phenomena were possibly attributed to the mechanical stress-induced interface trap generation and the enhanced hydrogen diffusion from atomic layer deposition-grown Al_2_O_3_ to IGZO under mechanical-bending stress during PBS. The subgap density of states was extracted before and after an application of PBS under both mechanical stress conditions. The obtained results in this study provided potent evidence supporting the mechanism suggested to explain the PBS-induced larger *∆V*_TH_s in both directions under mechanical-bending stress.

## 1. Introduction

Indium-gallium-zinc oxide (IGZO) thin-film transistor (TFT) is being widely used for the backplane of large-area active-matrix organic-light-emitting diode displays, owing to its excellent properties including high field-effect mobility (*μ*_FE_), low-off current, high uniformity, and low process temperature [1,2,3,4,5,6,7,8,9,10,11]. Recently, there is increasing interest in the application of IGZO TFTs in demonstrating the active-matrix backplane for flexible displays [12,13,14,15,16]. Flexible displays have many advantages over conventional glass substrate-based displays including better durability, lighter weight, and thinner dimension. In addition, flexible displays can enable a lot of new applications because of their ability to have unique curved shapes.

Nevertheless, there are still reliability problems that should be solved for practical applications of IGZO TFTs to the active-matrix backplane of flexible displays. As flexible displays can experience an external mechanical stress depending on the bending radius and direction, the IGZO TFTs can be subjected to the simultaneous mechanical and electrical stress when they are used for the backplane of flexible displays. However, unfortunately, there have been very few studies that examined the effect of simultaneous mechanical and electrical stress on the electrical properties of flexible IGZO TFTs, even though various studies were already conducted on the electrical or mechanical stress-induced performance degradation in IGZO TFTs [17,18,19,20,21,22]. In this work, we compared the positive-bias-stress (PBS)-induced instability of IGZO TFTs under mechanical-bending stress and no mechanical stress by using the flexible TFTs fabricated on the plastic substrate with an Al_2_O_3_ gate insulator deposited using the atomic layer deposition (ALD) technique. Our experimental results showed that the IGZO TFTs exhibited a threshold voltage shift (*∆V*_TH_) under an application of PBS with a turnaround behavior from the positive *∆V*_TH_ to the negative *∆V*_TH_ with an increase in the stress time regardless of an application of the mechanical stress. However, the magnitudes of *∆V*_TH_ in both the positive and negative directions exhibited larger values when a flexible IGZO TFT was under mechanical-bending stress than it was at the flat state. To find out the physical mechanism for the observed phenomenon, the subgap density of states (DOS) was extracted from the fabricated flexible IGZO TFTs before and after PBS application under mechanical-bending stress and no mechanical stress, respectively. The extracted values at each condition were correlated with larger magnitudes of PBS-induced *∆V*_TH_s in the IGZO TFT under mechanical-bending stress compared to that at the flat state.

## 2. Experimental Procedure

Figure 1a displays the schematic cross-sectional image of the flexible IGZO TFT fabricated on the polyethylene terephthalate (PET) substrate. The substrate was attached to a silicon wafer during the whole TFT fabrication process and separated from that when the TFT fabrication was completed. First, a 50-nm-thick SiO_2_ buffer layer was formed on the PET by e-beam evaporation. Next, a 20-nm-thick Cu film was deposited using e-beam evaporation and patterned to form the gate electrode. Then, a gate dielectric of 40-nm-thick Al_2_O_3_ was formed, using ALD at a low temperature of 80 °C, to avoid thermal damage on the PET substrate by utilizing Al(CH_3_)_3_ (trimethylaluminum—TMA) and water as precursors. A 35-nm-thick IGZO channel layer was formed by reactive sputtering using a polycrystalline IGZO target (In_2_O_3_:Ga_2_O_3_:ZnO = 1:1:1 mol %) at room temperature (RT). A 40-nm-thick Cu film was deposited and patterned to form the source/drain electrodes. Finally, the device was thermally annealed at 150 °C for 1 hour in air. Figure 1b,c display the photographic images of the fabricated IGZO TFT on the flexible PET substrate. 

The electrical parameters extracted from the representative device were as follows: A *V*_TH_ of 1.8 V; a *μ*_FE_ of 7.3 cm^2^/Vs; and a subthreshold swing of 0.32 V/dec. Here, *V*_TH_ was defined as the value of gate-to-source voltage (*V*_GS_) inducing the drain current (*I*_D_) of a width/length (*W*/*L*) × 10 nA at a drain-to-source voltage (*V*_DS_) of 5 V [23,24]. The mechanical stress was applied to the TFTs by using the customized bending plate with a ~20 mm bending radius (Figure 1d). The direction of bending was outward with respect to the flexible substrate (tensile stress) and parallel to the source–drain current path. The electrical properties of the devices were evaluated in the dark at RT using an Agilent 4156C precision semiconductor parameter analyzer. In addition, to remove the ambient effects on the experimental results [25,26,27], the electrical characterization was conducted in a 10 mTorr vacuum environment.

## 3. Results and Discussion

Figure 2a,b display the change of transfer curves as a function of the applied stress time under a positive *V*_GS_ of 8 V in flexible IGZO TFTs at the flat state and under the mechanical-bending stress, respectively. Measurements were made for TFTs with a *W*/*L* of 5 μm/20 μm at a *V*_DS_ of 5 V at RT. Figure 2a,b show that the IGZO TFTs exhibit *V*_TH_ turnaround characteristics under both mechanical stress conditions. *V*_TH_ shifts in the positive direction during the initial 500 s; but shifts in the negative direction after 500 s of stress. In previous works, the PBS-induced *V*_TH_ turnaround behavior was already observed in IGZO TFTs with a low-temperature ALD Al_2_O_3_ gate dielectric and was mainly ascribed to the effect of electron trapping and hydrogen release and diffusion [28]. The positive shift of *V*_TH_ at the initial stage of PBS was explained by the electron trapping in traps at the Al_2_O_3_/IGZO interface or bulk Al_2_O_3_, and the negative shift of *V*_TH_ after a long stress time was mainly attributed to the hydrogen diffusion from a low-temperature ALD Al_2_O_3_ gate dielectric into an IGZO layer. The hydrogen atom was assumed to be generated from the breakage of residual AlO-H bonds in the ALD Al_2_O_3_ by the energetic electrons in the TFT channel during PBS application [28]. When Al_2_O_3_ is deposited using ALD at low temperatures, the chemical reaction between AlO-H and TMA is less complete, which causes considerable AlO-H residues in Al_2_O_3_ [29]. As hydrogen is the effective donor in IGZO through the reaction of H^0^ + O^2−^→OH^−^ + e^−^ [30], the hydrogen doping shifts the *V*_TH_. of the IGZO TFT in the negative direction.

Figure 3 displays the schematic energy band diagram which illustrates the effects of electron trapping and hydrogen release and diffusion on the *V*_TH_ of IGZO TFTs. Figure 4 shows the *∆V*_TH_ versus stress time extracted from Figure 2a,b. From Figure 4, we can clearly observe that the magnitudes of *∆V*_TH_ in both the positive and negative directions exhibit larger values when a flexible IGZO TFT is subjected to the mechanical stress than when it is at the flat state. As far as we know, this is a phenomenon that has not been reported in the previous works. Considering that the IGZO TFT are subjected to the simultaneous mechanical and electrical stress when they are used for the backplane of flexible displays, it is very important to analyze the phenomenon observed in Figure 4.

To find out the physical mechanism for the observed phenomenon in Figure 4, the subgap DOS were extracted from the IGZO TFTs before and after PBS application under mechanical-bending stress and no mechanical stress, respectively, using the monochromatic photonic capacitance-voltage (*C-V*) technique [31]. Figure 5 displays the energy distribution of the subgap DOS obtained from the IGZO TFTs under mechanical-bending stress and no mechanical stress before PBS application, respectively. Figure 5 displays that the density of tail states near the conduction band edge (*E*_C_) extracted from the IGZO TFT under mechanical-bending stress are higher than that extracted from the TFT at the flat state, which is possibly ascribed to the larger number of structural defects at the interface in the mechanically-bended IGZO TFT due to different Young’s modulus values of Al_2_O_3_ (300 GPa) and IGZO (137 GPa) [27]. When the same strain is applied to each layer, the Al_2_O_3_ gate insulator undergoes larger stress compared to IGZO channel layer, which can generate the structural defects at the interface between Al_2_O_3_ and IGZO. As the interface trap states act as the electron trapping sites during PBS application in IGZO TFTs, it can explain the large positive shift of *V*_TH_ at the initial stage of PBS in the mechanically-bended IGZO TFT in Figure 4 [32]. From Figure 5, we can also observe that not only the density of tail states but the subgap DOS at 0.1–0.3 eV below *E*_C_ exhibits larger values in the mechanically-bended IGZO TFT than in the IGZO TFT at the flat state. In previous works, the mechanical strain was reported to generate the oxygen vacancies in IGZO, including ionized ones [20,21]. Considering that the increase in the donorlike states at ~*E*_C_–(0.1–0.3 eV) is most likely to result from the increase in the ionized oxygen vacancies in IGZO [33,34], an increase in the subgap DOS at 0.1–0.3 eV below *E*_C_ in Figure 5 can be possibly ascribed to the increased oxygen vacancies in IGZO due to the mechanical bending stress.

Figure 6a,b compares the subgap DOS extracted from the IGZO TFTs before and after PBS application (stress time: 2000 s) at the flat state and under the mechanical-bending stress, respectively. Figure 6 shows that the subgap DOS near *E*_C_ increases after PBS application under both mechanical conditions. However, it increases more pronouncely after PBS in the IGZO TFT under mechanical bending stress than that at the flat state. In previous reports, the negative shift of *V*_TH_ after a long stress time was mainly attributed to the hydrogen diffusion from a low-temperature ALD Al_2_O_3_ gate dielectric into an IGZO layer [28]. As the hydrogen acts as an effective donor in IGZO, it generates the donor states near *E*_C_ and shifts the *V*_TH_ of the IGZO TFT in the negative direction. The PBS-induced increase of subgap DOS near *E*_C_ observed in Figure 6 is possibly ascribed to the increase of the hydrogen concentration inside the IGZO after PBS application. In addition, a more pronounced increase of subgap DOS near *E*_C_ in the IGZO TFT under mechanical-bending stress is considered as a result of more enhanced hydrogen diffusion from Al_2_O_3_ gate dielectric into an IGZO layer in the mechanically stressed IGZO TFT. This result is consistent with that in Figure 4, which shows a more pronounced negative shift of *V*_TH_ after a long stress time in the mechanically stressed IGZO TFT. More enhanced hydrogen diffusion in the mechanically stressed IGZO TFT is believed to be caused from the increased oxygen vacancies which can act as the hydrogen hopping site in IGZO. Figure 7 displays the schematic diagram illustrating the mechanism responsible for more enhanced hydrogen diffusion form ALD Al_2_O_3_ gate dielectric into an IGZO layer in IGZO TFTs under mechanical stress.

## 4. Conclusions

In this research, we compared the PBS-induced instability of IGZO TFTs under the mechanical-bending stress and no mechanical stress by using the flexible IGZO TFTs fabricated with a low-temperature ALD Al_2_O_3_ gate dielectric. The IGZO TFTs exhibited a *∆V*_TH_ under an application of PBS with a turnaround behavior from the positive *∆V*_TH_ to the negative *∆V*_TH_ under both mechanical stress conditions. The magnitudes of *∆V*_TH_ in both directions exhibited higher values when a flexible IGZO TFT was mechanically bended than it was at the flat state. The observed phenomena were considered as a result of the additional interface trap generation and the enhanced hydrogen diffusion from ALD Al_2_O_3_ to IGZO by the mechanical-bending stress during an application of PBS. The subgap DOS was extracted before and after an application of PBS under both mechanical stress conditions using the monochromatic photonic *C-V* technique. The obtained results were correlated with a PBS-induced larger *∆V*_TH_s in both directions under mechanical-bending stress as compared to at the flat state. To fully understand the degradation mechanism, a further study that compares the effects of simultaneous mechanical and electrical stress in flexible IGZO TFTs having Al_2_O_3_ gate dielectrics with different hydrogen contents needs to be conducted in the future.

## Figures and Tables

**Figure 1 materials-12-03248-f001:**
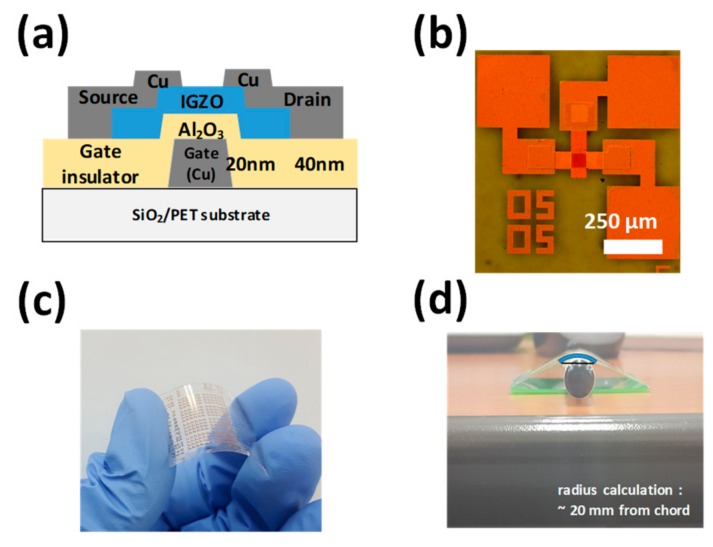
(**a**) Schematic cross-sectional image of the flexible indium-gallium-zinc oxide (IGZO) thin-film transistor (TFT) fabricated on the polyethylene terephthalate (PET) substrate. (**b**), (**c**) Photographic images of the fabricated flexible IGZO TFT. (**d**) Photographic image of the customized bending plate with a ~20 mm bending radius.

**Figure 2 materials-12-03248-f002:**
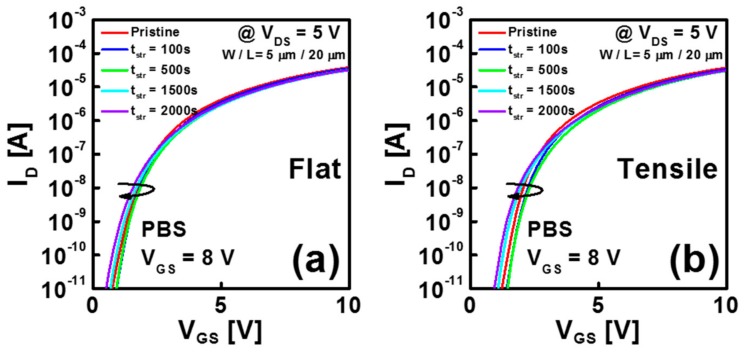
Change of transfer curves as a function of the applied stress time under a *V*_GS_ stress of 8 V in flexible IGZO TFTs (**a**) at the flat state and (**b**) under the mechanical-bending stress.

**Figure 3 materials-12-03248-f003:**
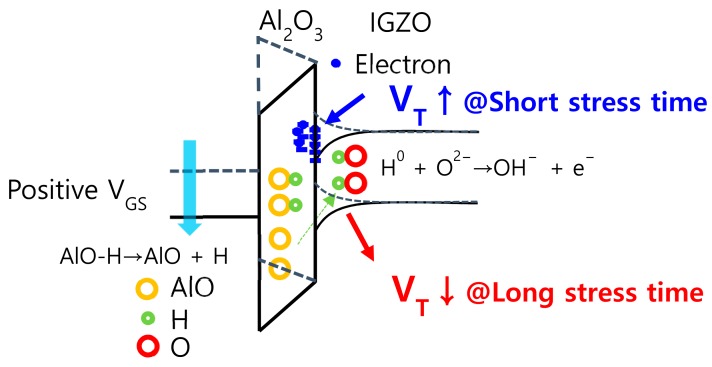
Schematic energy band diagram which illustrates the effects of electron trapping and hydrogen release and diffusion on the *V*_TH_ of IGZO TFTs.

**Figure 4 materials-12-03248-f004:**
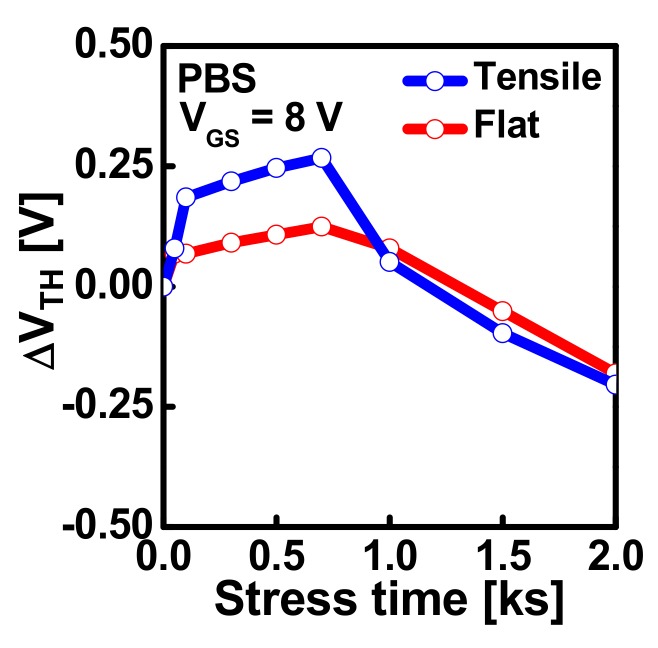
*∆V*_TH_ versus stress time under a *V*_GS_ stress of 8 V in flexible IGZO TFTs at the flat state and under tensile bending stress.

**Figure 5 materials-12-03248-f005:**
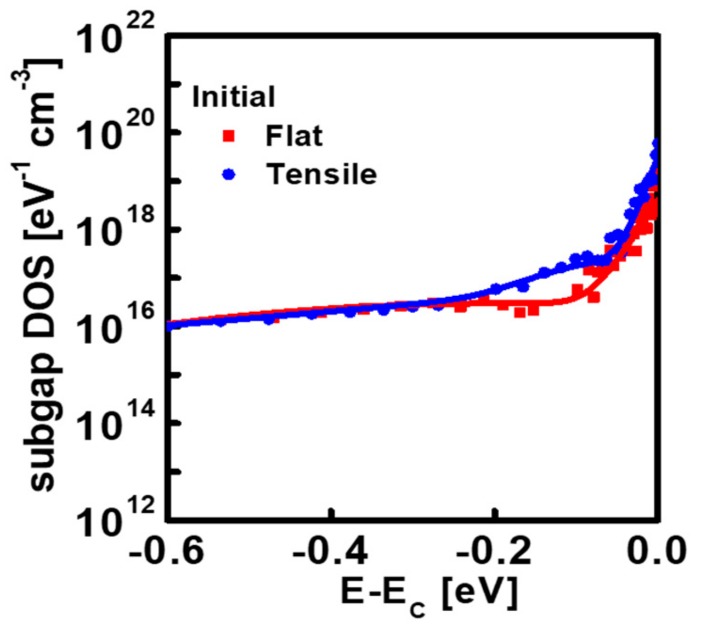
Energy distribution of the subgap density of states (DOS) obtained from the IGZO TFTs before positive-bias-stress (PBS) application under the tensile bending stress and no mechanical stress.

**Figure 6 materials-12-03248-f006:**
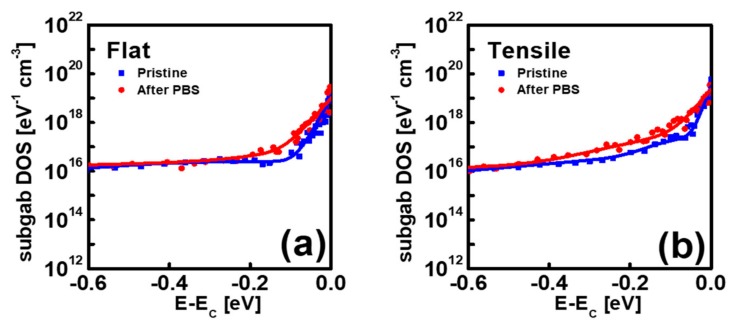
Energy distribution of the subgap DOS obtained from the IGZO TFTs before and after PBS application (stress time: 2000 s) (**a**) at the flat state and (**b**) under tensile bending stress.

**Figure 7 materials-12-03248-f007:**
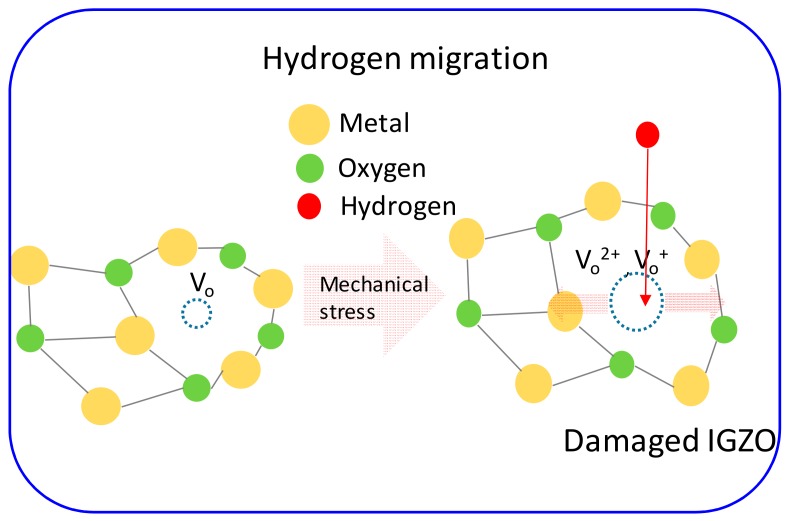
Schematic diagram illustrating the physical mechanism responsible for more enhanced hydrogen diffusion form atomic layer deposition (ALD) Al_2_O_3_ gate dielectric into an IGZO layer in IGZO TFTs under mechanical bending stress.

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
