# Peer review of "Effect of Simultaneous Mechanical and Electrical Stress on the Electrical Performance of Flexible In-Ga-Zn-O Thin-Film Transistors"

_materials, 2019, doi:10.3390/ma12193248_

Round 1

Reviewer 1 Report

Effect of simultaneous mechanical and electrical stress on the electrical performance of flexible In-Ga-Zn-O thin-film transistors

 Youngjin Seo et al.

The subject of the article is interesting, as there is nowadays an important development of electronic devices on flexible displays. The authors address the effect of simultaneous mechanical and electrical stress on the electrical performances of the IGZO TFTs. These stresses have definitely to be investigated on this kind of devices. The presented work is interesting. Anyhow, a few corrections have to be done:

- L. 91-92: It is not clear why the hydrogen plays such an important role. Even though the authors give a reference [23], the authors should mention shortly the fabrication route of Al2O3. The authors should mention at l. 69-70 the fabrication route indicated in ref 23 “Low-temperature ALD was applied for depositing the Al2O3 gate dielectric by using Al(CH3)3 (trimethylaluminum; TMA) and water as the precursors.”

- l. 136 add : The link between localized deep levels in the upperhalf of the band gap (slightly below the conduction band) and the local oxygen deficiencies situated nearby one or several metal atoms was experimentally put in evidence [Revenant].

[Revenant] Revenant, C.; Benwadih, M.; Proux, O. Local structure around Zn and Ga in solution‐processed In–Ga–Zn–O and implications for electronic properties. Phys. Status Solidi RRL 2015, 9, 652, https://doi.org/10.1002/pssr.201510322

This will give more strength to your assertion at l. 137-138.

 - The authors conclude that the observed phenomena are possibly attributed to the mechanical stress-induced interface trap generation and the enhanced hydrogen diffusion from atomic layer deposition-grown Al2O3 to IGZO under mechanical-bending stress during PBS. The authors could also mention that in the future, other Al2O3 deposition modes could be explored (with no or at least less hydrogen left).

- l. 41: comfortable ? It looks strange.

Write: « In addition, flexible displays can enable a variety of new applications because of their ability to have unique curved shapes. »

- In Fig. 1(a), it is indicated 20 mm, 40 mm. It should be 20 nm, 40 nm. The authors wrote in the text a 20-nm-thick Cu film and 40-nm-thick Al2O3.

- l. 106 : x 10 nA Where does it come from ?

- Fig. 4: Remove the two purple arrows with Vth up and down as the graph is clear enough.

Reviewer 2 Report

This manuscript investigates the effect of simultaneous mechanical and electrical stress on the electrical performance of flexible IGZO TFTs. The obtained results should benefit the actual applications of flexible displays. However, the discussion about the physical mechanisms in this manuscript is still not very clear. Some key points are summarized as following:

There were no passivation layers for the IGZO-TFT devices used in this study. So, the influence of ambient conditions on the bias-stress stability and mechanical-bending stability cannot be ignored. During PBS tests, the hydrogen atoms diffused from GI into IGZO. Why was that? If the tensile stress enhanced this hydrogen diffusion, the threshold voltage shifts during PBS tests should become smaller (because the electrons dominated by hydrogen atoms make the transfer curves shift negatively). But the experimental data (as shown in Fig. 2) did not support this assumption. Why? Why did the authors choose the bias-stress voltage of 8 V? This value is seldom used in PBS tests. Please confirm the channel parameters listed in the manuscript (W/L=5um/20um).

Round 2

Reviewer 2 Report

No comment.

Author Response

Thanks for the reviewer's comments.